# Identification of key regulators in pancreatic ductal adenocarcinoma using network theoretical approach

**Kankana Bhattacharjee, Aryya Ghosh** *

Department of Chemistry, Ashoka University, Sonipat, Haryana, India

* aryya.ghosh@ashoka.edu.in

## Abstract

Pancreatic Ductal Adenocarcinoma (PDAC) is a devastating disease with poor clinical outcomes, which is mainly because of delayed disease detection, resistance to chemotherapy, and lack of specific targeted therapies. The disease's development involves complex interactions among immunological, genetic, and environmental factors, yet its molecular mechanism remains elusive. A major challenge in understanding PDAC etiology lies in unraveling the genetic profiling that governs the PDAC network. To address this, we examined the gene expression profile of PDAC and compared it with that of healthy controls, identifying differentially expressed genes (DEGs). These DEGs formed the basis for constructing the PDAC protein interaction network, and their network topological properties were calculated. It was found that the PDAC network self-organizes into a scale-free fractal state with weakly hierarchical organization. Newman and Girvan's algorithm (leading eigenvector (LEV) method) of community detection enumerated four communities leading to at least one motif defined by G (3,3). Our analysis revealed 33 key regulators were predominantly enriched in neuroactive ligand-receptor interaction, Cell adhesion molecules, Leukocyte transendothelial migration pathways; positive regulation of cell proliferation, positive regulation of protein kinase B signaling biological functions; G-protein beta-subunit binding, receptor binding molecular functions etc. Transcription Factor and mi-RNA of the key regulators were obtained. Recognizing the therapeutic potential and biomarker significance of PDAC Key regulators, we also identified approved drugs for specific genes. However, it is imperative to subject Key regulators to experimental validation to establish their efficacy in the context of PDAC.

## Introduction

Pancreatic ductal adenocarcinoma (PDAC) is a common type of cancer originating from the pancreatic glands and is characterized by a rapidly progressive course and a dismal prognosis [1]. Combination strategies targeting multiple signaling pathways supporting tumor growth and propagation are active areas of contemporary research that will likely transform the treatment paradigm of pancreatic cancer [2]. This is a highly detrimental disease with dismal

**Data Availability Statement:** All the RNA-seq data are available from the Gene Expression Omnibus repository GSE171485 (https://www.ncbi.nlm.nih.gov/geo/query/acc.cgi?acc=GSE171485).

**Funding:** Indian Council of Medical Research (ICMR) (BMI/11(92)/2022) The funders had no role in study design, data collection and analysis, decision to publish, or preparation of the manuscript.

**Competing interests:** No

clinical outcomes, primarily attributable to delayed disease detection, chemotherapy resistance, and absence of specific targeted therapies. The identification of novel therapeutic targets and/or early biomarkers for the disease has the potential to significantly improve the clinical management of PDAC and extend the lives of patients. To develop successful drug therapies, a deeper understanding of the molecular mechanisms underlying drug targeting is essential. A connectivity Map is a platform that provides information on the signaling pathways activated by a particular drug and can serve as a valuable resource for drug development. A promising strategy for treating PDAC involves exploiting aberrant metabolic processes in cancer cells, particularly PDAC cells. Cancer cells alter their metabolic pathways, a process regulated by complex and poorly defined interplay between intrinsic and extrinsic factors.

The incidence of PDAC is rising, and it is now one of the leading causes of cancer-related deaths in developed countries [3]. Current treatment paradigms for PDAC, including targeted therapy, have shown limited success in improving survival outcomes [4]. The molecular heterogeneity of PDAC and its complex tumor microenvironment contribute to its resistance to conventional treatments and immunotherapies [5]. Retinoids, such as retinoic acid (RA), have shown potential in maintaining normal pancreatic functions and have been explored as a therapeutic option for PDAC [6]. The use of predictive molecular markers and cancer gene panel testing may help in selecting personalized therapies for PDAC patients [7]. Accurate diagnosis of PDAC is crucial, as it can be easily misdiagnosed as other pancreatic neoplasms, such as acinar cell carcinoma (ACC) or neuroendocrine tumor (PNET). Further research is needed to overcome the challenges in PDAC management and improve patient outcomes. Recent research is exploring a wide range of novel therapeutic targets for PDAC, including genomic alterations, tumor microenvironment, and tumor metabolism [8]. The rapid evolution of tumor genome sequencing technologies paves the way for personalized, targeted therapies for PDAC [9]. Immunotherapy with immune checkpoint inhibitors has shown promise in PDAC, particularly in tumors harboring mismatch repair deficiency (dMMR) and high microsatellite instability (MSI-H) [4]. The presence of a dense fibrotic stroma in PDAC creates a physical barrier around the cancer cells, hindering drug delivery and promoting tumor growth and treatment resistance [10]. Advances in surgical approaches, immunotherapeutic approaches, and targeted therapies are being explored to overcome the treatment refractory nature of PDAC and improve patient outcomes [11]. The tumor microenvironment (TME) plays a crucial role in PDAC development, progression, and treatment resistance [12]. Immunotherapy, including checkpoint inhibition and immune-based therapies, has shown promise in PDAC treatment [13]. Additionally, targeted therapies that focus on specific signaling pathways and components of the TME, such as fibro-blasts and cancer-associated fibroblasts, are being explored.

PDAC is an aggressive and deadly cancer with limited treatment options. Current standard of care treatments for advanced PDAC include systemic chemotherapy regimens such as gemcitabine/nab-paclitaxel and FOLFIRINOX, which have improved clinical outcomes [14]. However, the 5-year survival rate for PDAC remains low, highlighting the need for new therapies [15]. The tumor microenvironment (TME) of PDAC plays a significant role in tumorigenesis and may contain promising novel targets for therapy [16]. Adjuvant chemotherapy has been shown to offer a survival advantage over surgery alone in resected PDAC [17]. However, the addition of chemoradiation therapy (CRT) to chemotherapy does not provide a survival advantage and is not recommended [18]. Surgical resection is the first choice for treatment of pancreatic acinar cell carcinoma (PACC), but there is no standard treatment option for inoperable disease. Understanding the risk factors for PDAC can help in screening and counseling patients for lifestyle modifications.

A disorder typically arises from disruptions within the intricate internal web of connections [19] among genes that serve related functions, rather than from the abnormality of a single gene. This resulted in the adoption of a systemic approach to biological issues, founded on the principle that comprehending the involvement of different genes/proteins is essential [20] in disease initiation and progression. It is necessary to consider the entire network of interactions within a living system [21]. In this context, it is important to employ Network Medicine [22], as this method aids in the investigation of disease pathways and modules linked to complex illnesses.

In this study, the protein interaction maps are analyzed through graph/network theory to get insights about the theoretical aspect of complex networks [23]. According to the graph theory, analysis of the topological structure of a network (PPI network in our study) provides important information of the network [24] through which novel disease genes and pathways, biomarkers and drug targets for complex diseases can be identified [25].

Thus, the focus of our study is on the protein-protein interactions network/graph of PDAC, constructed from differentially expressed genes with an aim to understand the architectural principle of the network/graph (random, small world, scale-free or hierarchical). We further extended our study to the prediction of important key regulators of the network which have fundamental importance due to their activities and regulating mechanisms in the network [26]. It is expected that the findings of this study will advance our understanding of the initiation and progression of PDAC, thereby, strengthening different therapeutic approaches for PDAC.

This work was to establish an unbiased catalogue of change in (up and down regulation) gene expression for the PDAC samples. we bestowed 12 samples of PDAC analysis of gene-expression from normal and cancer patients. Finally, the comparative study of these conditions (PDAC Cancer and Normal) has affirmed the thirty-three (33) prominent genes. These are intricated in basic functioning of the cell like as in the rearrangement of the cytoskeleton, tissues development and activation of immune system. This work will be useful for the researchers to collect evidences against PDAC working in in-vitro.

## Materials and methods

### Preprocessing and acquisition of dataset

The RNA-seq dataset GSE171485 [27] was downloaded from the NCBI GEO database [https://www.ncbi.nlm.nih.gov/].

### Data analysis and visualization of differentially expressed genes

The data analyses were conducted using R Studio (4.2.2) and R. The fold change method was employed to identify differentially expressed genes. The fold change for a gene is calculated by subtracting the intensities measured in two samples (control vs clinical groups). This value is referred to as the fold change. These raw values are typically log transformed (usually log2) [28]. Another method to calculate the fold change ratio involves dividing the two measured intensities for a given gene in two samples. A change of at least two-fold (up or down regulated) considered to be significant.

### Screening of DEGs

We utilized the LIMMA package [29] to analyze the data and identify Differentially expressed genes (DEGs) by assessing gene expression values. This approach employs Linear Modeling (LM) [30] and Empirical Bayes (EB) [31] techniques to perform f-tests [32] and t-tests [33],

while simultaneously reducing standard errors. The objective of this method is to produce results that are dependable and reproducible, thereby enhancing the precision and reliability of the statistical analyses. DEG analysis was conducted to compare gene expression levels between control and clinical groups. The R package "ggplot2" [34] was employed for data visualization purposes. The fold change criteria for each gene were established based on a Benjamin-Hochberg adjusted p-value threshold of 0.05 and a significance level of $p \leq 0.05$.

## Construction of protein-protein interaction network of PDAC

The construction of the PDAC PPI (Protein-Protein Interaction) network was undertaken using the STRING database (The Search Tool for the Retrieval of Interacting Genes, (<http://string-db.org/>) with an interaction score threshold of $> 0.4$. This approach enables the exploration and analysis of protein-protein interactions, which can be either physical or functional associations. These associations are derived from text-mining of literature, co-expression analysis, genomics context-based predictions, computational predictions, and high-throughput experimental data, as well as the aggregation of previous knowledge from other databases. The network was subsequently visualized and analyzed using the Cytoscape software (version 3.6.1) [35].

## Gene ontology (GO) enrichment analysis

Gene ontology terms provide a controlled vocabulary that is divided into three categories: Molecular Function, Biological Process, and Cellular Location. To conduct a preliminary investigation into the functional differences of DEGs, they were submitted to DAVID (Database for Annotation, Visualization and Integrated Discovery), an online software (<http://david.abcc.ncifcrf.gov/home.jsp>), to enrich the set of DEGs with possible GO terms [36].

## Delineation of global network topological properties

The structure of the network can be examined through its topological properties, which establish the connections between nodes and illustrate their interactions. The topology of the network is defined by the probability of degree distributions ($P(k)$), clustering coefficient ($C(k)$), and neighborhood connectivity ($C_N(k)$), which exhibit a power law distribution. This adherence to the power law distribution indicates the presence or absence of scale-free properties in the network. The network constructed for PDAC using differentially expressed genes extracted from the RNA-seq dataset GSE171485 follows the power law distribution in its probability of degree distributions (P(k)), clustering coefficient (C(k)), and neighborhood connectivity ($C_N(k)$). This is depicted in "**Fig 4**" and confirmed by the matrix representation of all properties [37].

The network's behavior is characterized by equations (see Eqs 1, 2, 3) that reveal a hierarchical structure in synergy with the scale-free nature of the network. The power law distribution was fitted to the topological properties of the network using a standard statistical fitting procedure.

$$P(k) \sim k^{-\alpha} \tag{1}$$

$$C(k) \sim k^{-\beta} \tag{2}$$

$$C_N(k) \sim k^{\emptyset} \tag{3}$$

The positive value in phi of connectivity parameter shows assortative nature of the network.

While, the negative value in alpha (α) of degree distribution shows availability of each node in the network. The negative value in beta of clustering parameter shows disassortative in the communication between the nodes in network [23].

## Communities finding: LEV method

To investigate the nature and topological properties of hierarchical networks at various levels of organization is beneficial for understanding network behavior and uncovering the organizing principles that govern them. There are several methods for detecting communities within these networks, one of which is the leading eigenvector (LEV) method. This method calculates the eigenvalue for each edge, giving greater importance to links rather than nodes. For this study, we applied the LEV detection method using the 'igraph' [38] package in R. We used this code to detect modules in the main network, sub-modules within each module at different levels of organization, and so on, ultimately identifying motifs (i.e. 3 nodes and 3 edges) [39]. Throughout this process, we adhered to the criterion of identifying any sub-module as a community if it contained at least one motif (defined by G (3,3)).

## TF/miRNAs screening

To identify miRNAs targeting key signature genes, we employed MIENTURNET (MicroRNA ENrichment TURned NETwork). MIENTURNET [40] is an interactive web application designed for micro-RNA target enrichment analysis, primarily relying on the TargetScan program for sequence-based miRNAs target predictions [41]. By utilizing the MIENTURNET software, we successfully achieved significant functional enrichment of the predicted miRNAs.

## Drug gene interaction analysis

To identify potential drugs for treating PDAC, we utilized the DGIdb web tool [42] a database of drug-gene interactions and druggable genes.

## Results

This work provides information in RNA-seq dataset GSE171485 on the structure and function of interacting genes. The results of the differential expression analysis, as shown in volcano plot "**Fig 1**", indicate that 772 DEGs were identified. Of these, 341 genes were found to be down-regulated and 431 were up-regulated, with the threshold cut-offs set at log2 |FC| > = 1, P-value < 0.05, and Padj ≤ 0.05.

To get the idea of what could be the effect of DEGs can only be possible when we have some preliminary insight in to their individual function. As the term gene ontology (GO) enrichment gives us the opportunity to get the basic idea about any gene. GO enrichment analysis of significantly enriched DEGs between disease and control was categorized into biological process, cellular components and molecular function. Among these down-regulated DEGs, potassium ion transport, cell-cell signaling, neuropeptide signaling pathway were most significantly down-regulated biological process in disease "Fig 2A". While other important biological process associated with these down-regulated DEGs were muscle contraction, cellular response to zinc ion, response to hypoxia, negative regulation of cytosolic calcium ion concentration and positive regulation of apoptotic process. While most significantly down-regulated cellular components and molecular function associated with the DEGs were extracellular region, mitochondrion, potassium channel complex, endoplasmic reticulum, mitochondrion inner membrane and protein binding, methyltransferase activity, structural constituent of ribosome respectively "Fig 2B and 2C". KEGG pathways enrichment of down-regulated genes were

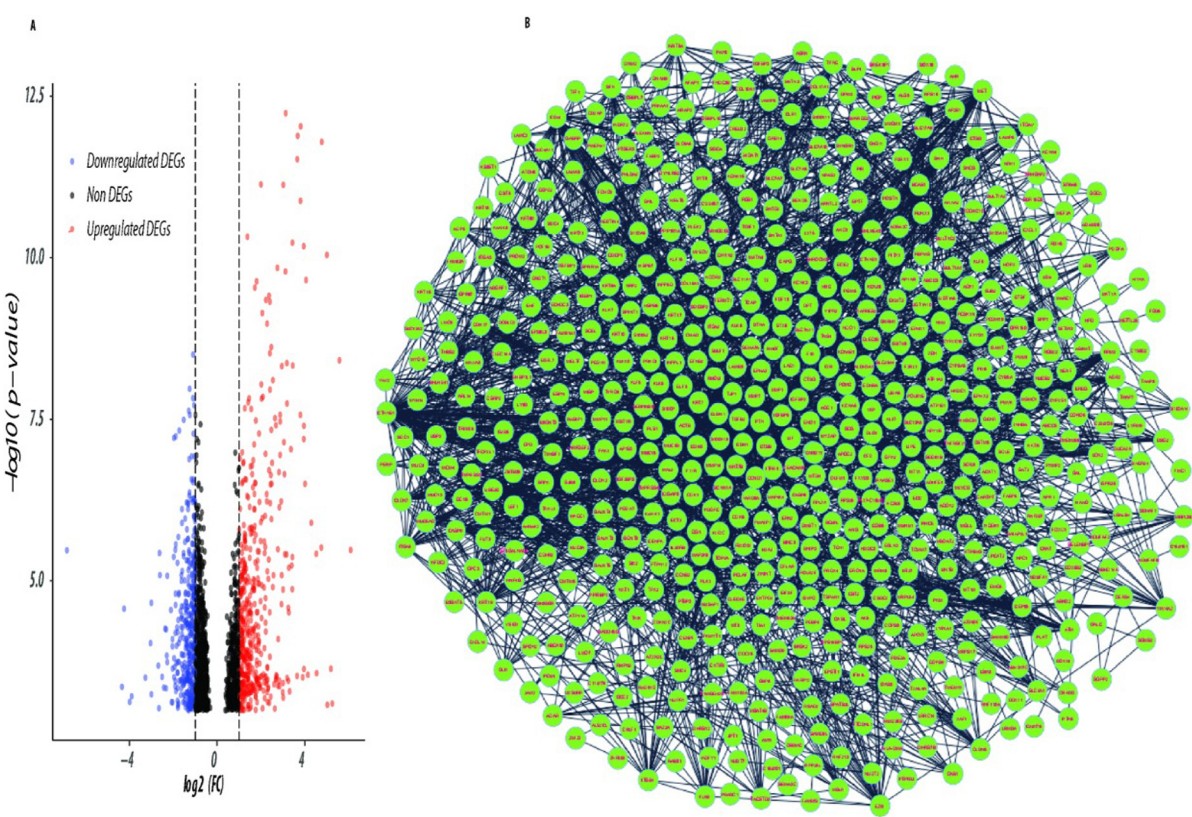

**Fig 1.** Figure showing (A) volcano plot showing up & down regulated expressed genes (B) Network of differentially expressed genes.

enriched in metabolic pathway, peroxisome cAMP signaling pathway and chemical carcinogenesis-reactive oxygen species "Fig 2D".

Among the top 15 up-regulated DEGs, cell-cell adhesion, positive regulation of cell migration, extracellular matrix organization, tissue development, epithelial cell differentiation and integrin-mediated signaling pathway were most significantly up-regulated biological process in PDAC "Fig 3A". GO classification indicates the up-regulation of calcium ion binding, calcium-dependent protein binding, cadherin binding, virus receptor activity and insulin-like growth factor I binding related molecular function in PDAC "Fig 3B".

While most significantly up-regulated cellular components were plasma membrane, extracellular exosome, and perinuclear region of cytoplasm Figure as shown in "Fig 3C". KEGG pathways were enriched in ECM-receptor interaction, p53 signaling pathway, pathways in cancer and proteoglycan in cancer "Fig 3D".

## PDAC network architecture reveals hierarchical scale-free features

To gain insights into the PPI network of PDAC's structural features, we analyzed its topology, specifically the probability of degree distribution $P(k)$, clustering coefficient $C(k)$ neighborhood connectivity $C_N(k)$, and centrality measurements. We employed the statistical fitting technique proposed by Clauset et al. [43] to verify that the graph's architecture follows power-law behavior. Our results indicate that all statistical p-values, calculated against 2500 random samplings, are greater than the critical value of 0.1, and the goodness of fits are less than or equal to 0.35. The data points of all the topological parameters fit power law when plotted against the degree k of the PDAC network.

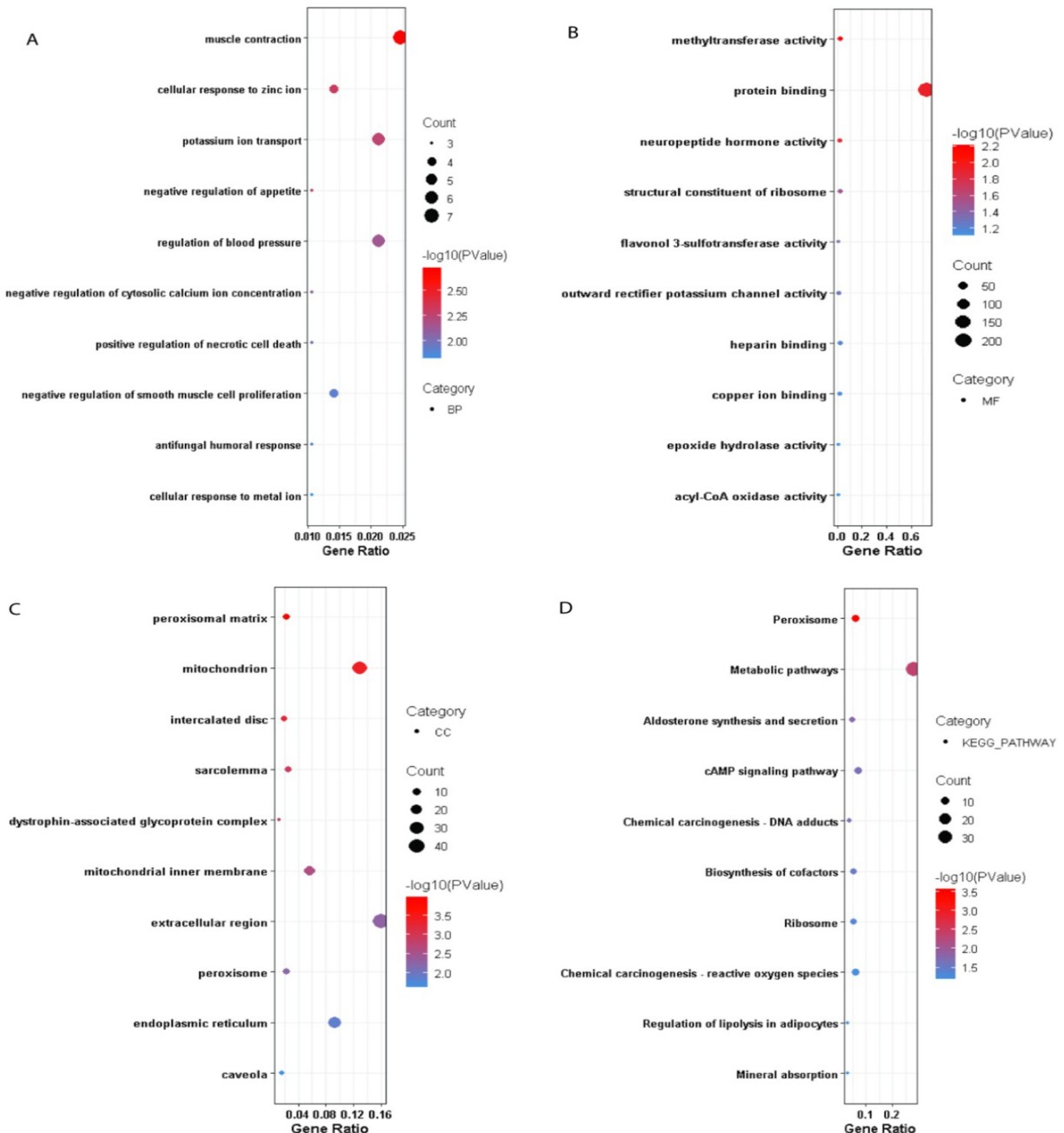

**Fig 2. Gene ontology analysis for down-regulated genes of pancreatic ductal adenocarcinoma.**

The values of the power-law exponents for each of the topological properties of the complete network were calculated:

$$\begin{pmatrix} P \\ C \\ C_N \end{pmatrix} \sim \begin{pmatrix} K^{-\alpha} \\ K^{-\beta} \\ K^{+\emptyset} \end{pmatrix}; \begin{pmatrix} \alpha_0 \\ \beta_0 \\ \emptyset_0 \end{pmatrix} \rightarrow \begin{pmatrix} 0.294 \\ 0.165 \\ 0.122 \end{pmatrix}$$

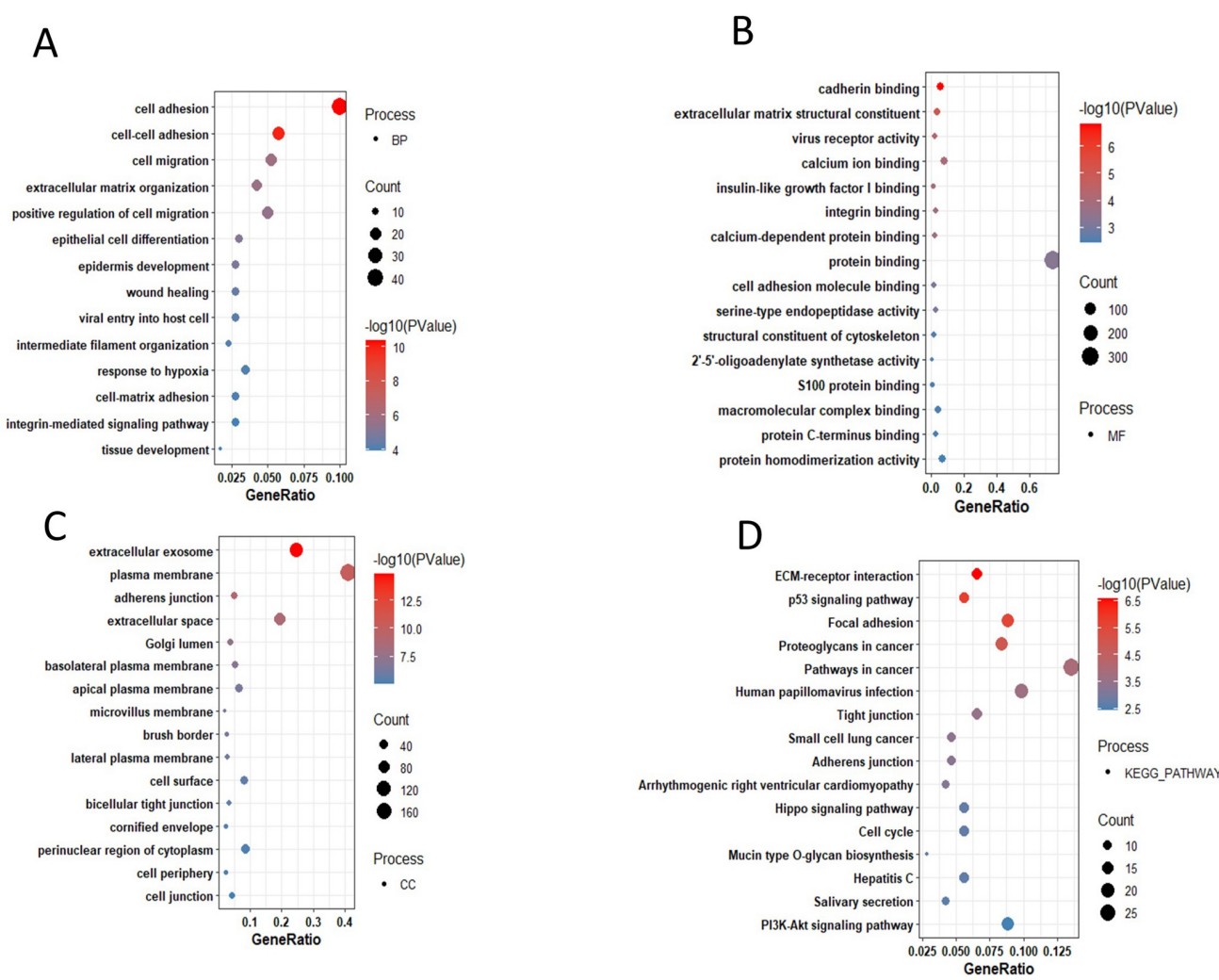

**Fig 3. Gene ontology analysis for up-regulated genes of pancreatic ductal adenocarcinoma.**

The negative values of α = 0.294; (α<2) and β = 0.165; (β<1) suggest the hierarchical nature of the PD network, indicating the existence of well-defined successive interconnected communities with sparsely distributed hubs in the network. The values of the exponents of $P(k)$, $C(k)$, and $C_N(k)$($\emptyset = 0.122$; ($+\emptyset \leq 0.5$)) suggest that the network, though not strongly hierarchical, falls into the category of a weak hierarchical scale-free network. The negative value of β indicates that as k increases, $C(k)$ decreases, suggesting that nodes with a high degree have a low tendency to cluster, further indicating a hierarchy of hubs, in which the most densely connected hub is linked to a small fraction of all other nodes. The power-law distribution observed in $P(k)$ is indicative of the scale-free nature of the small-world network, with the negative value of α indicating that a small number of nodes possess a high degree while the majority of nodes have a low degree, which is consistent with the network's scale-free behavior. The positive value of $\emptyset$ suggests that the network exhibits assortative mixing, where edges predominantly connect heavily connected nodes, regulating the system's behavior.

Further,

$$\begin{pmatrix} C_B \\ C_C \\ C_E \end{pmatrix} \sim \begin{pmatrix} K^\gamma \\ K^\delta \\ K^\psi \end{pmatrix}; \begin{pmatrix} \gamma_0 \\ \delta_0 \\ \Psi_0 \end{pmatrix} \rightarrow \begin{pmatrix} 0.302 \\ 0.095 \\ 1.008 \end{pmatrix}$$

The positive values of the exponents γ, δ, and ψ of the three distributions $C_B(k)$, $C_c(k)$ and $C_E(k)$ respectively, indicate that the network exhibits hierarchical scale-free or fractal features. These positive values signify that $C_B(k)$, $C_c(k)$ and $C_E(k)$ increase as the degree k increases when plotted against it "Fig 4C, 4D and 4F". The increasing value of $C_B(k)$ as k increases suggests that nodes with a high degree have high $C_B(k)$, indicating that these larger hubs have a major influence on the information transmission in the network compared to nodes with a low degree. Similarly, the direct proportionality between $C_c(k)$ and k suggests that high-degree nodes are quick spreaders of information in the network, indicating their high $C_c(k)$. The positive value of ψ indicates that nodes with a high degree have high $C_E(k)$ as well, suggesting their influence in the network due to their ability to spread information. The positive value of ψ also signifies the connectedness between high-degree nodes, which is in agreement with the assortative mixing in the network.

Through a meticulous study of these topological properties, it was found that the PDAC network self-organizes into a scale-free fractal state with weakly hierarchical organization.

**Key regulators uncovered through clustering and tracing.** Owing to the significance of regulator genes(RGs) as functional bottlenecks in the initiation and progression of a disease by regulating the expression of a plethora of downstream effector genes [44], we identified the most potent RGs of the PDAC network. Newman and Girvan's algorithm helped us untangle the PDAC network and the network was observed to be organized in five hierarchical levels

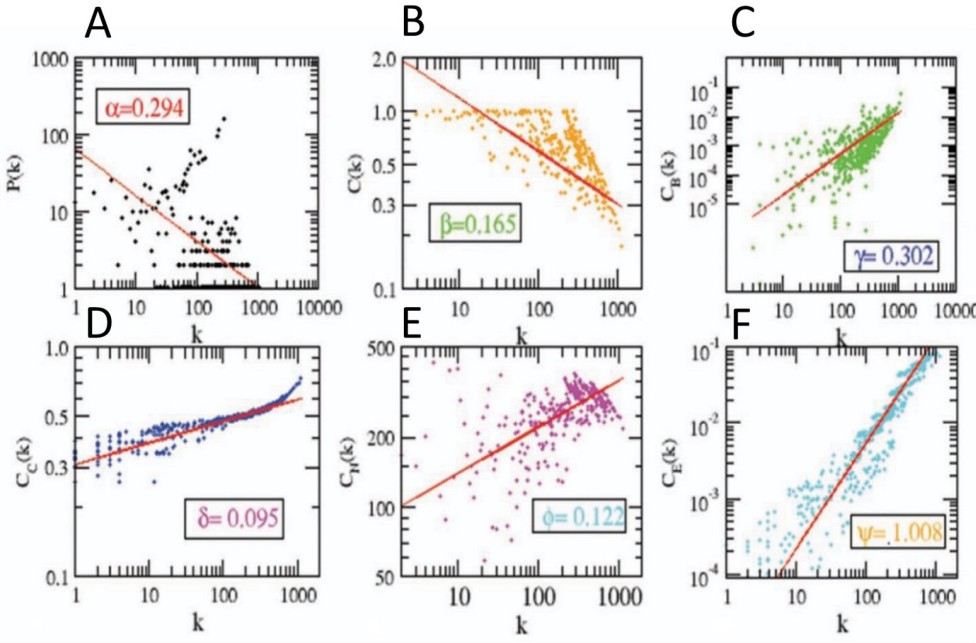

**Fig 4. Topological properties of Network with power law fit. (A)** Probability of node degree distribution. **(B)** Clustering coefficient vs degree distribution. **(C)** Betweenness centrality vs degree distribution. **(D)** Closeness centrality vs degree distribution **(E)** Neighborhood connectivity vs degree distribution. **(F)** Eigenvector value vs degree distribution.

using this algorithm "Fig 5". After tracing of the G(3,3) triangular structure genes from top to bottom organization through these levels of hierarchy, 33 genes were revealed to be the RGs of the PDAC network, the criterion being their presence at every topological level "Fig 5". This agrees with the definition of RGs, according to which, RGs are the genes/proteins which are deeply rooted from top to bottom organization of the network. These RGs are the backbone in maintaining a network's stability as they capacitate the network to combat any unacceptable alterations in it.

RNF213, EPSTI1 and XAF1 separated their way from the rest of the RGs from the first level itself and then took the path hand in hand till the motif level. Whereas, GAL, VIP, GNRH1, NMU, VIPR2, and GNG11 RGs all moved into a different sub-module of the first level and took path at the motif level. Genes MMP1, CTSG, F2RL1, F10, PLAT, F2R, PDGFA, FGF13, PTN, APOC2, SDC1, and SDC4 moved into same module at level 2. Further, genes MMP1, CTSG, F2RL1, F10, PLAT, and F2R makes sub-module and belongs to same sub-module till level 4 and separated at level 5 as triangular motif structure "Table 1".

In the third level, PDGFA, FGF13, PTN, APOC2, SDC1, and SDC4 moved into the same sub-module and got separated at level four into two sub-modules. Afterward, these RGs moved separately till they reached the motif level i.e., the 5th level. Genes AP2B1, AP1S3 DNM2, EGFL7, RASIP1, SOX18, CLDN2, CDH3, TJP1, CLDN5 and PCDH1 clustered in same sub-module at level 2 and got separated at levels 3, 4 and reached motif levels "Fig 6".

## Emergence of low degree node accompanied by high degree node as key regulators

In a network, when a node's degree is low, the node gains what strength it has from its neighbors and thus the influence it has over the network is a function of its neighboring degree. Whereas for high degree nodes, the strength of the nodes comes from their large number of connections rather than their neighboring degree [45]. In addition, a low degree bridge node, connecting two high degree nodes, is very important in a network despite its lower degree [46]. Thus, a node's degree is not the sole determinant of its essentiality, rather, it depends on the topological position of that node. This is reflected in our results, where, F2RL1, F10, APOC2, CLDN2, PCDH1 which has quite a low degree (4,4,8,11,3) respectively in the primary network, found out to be RGs based on its ability to make it to the last level of the organization. F2RL1 formed a motif in the last level of the organization with another KR, MMP1, which has a fairly high degree (36) in the primary network. Gene F10 formed a motif with PLAT having degree 17, APOC2 formed motif with SDC1 having degree 40 and PCDH1 formed motif with TJP1 which fairly high degree is [42] in primary network. This shows that a low degree node i.e., F2RL1, F10, APOC2, CLDN2, and PCDH1 are also an important information propagator

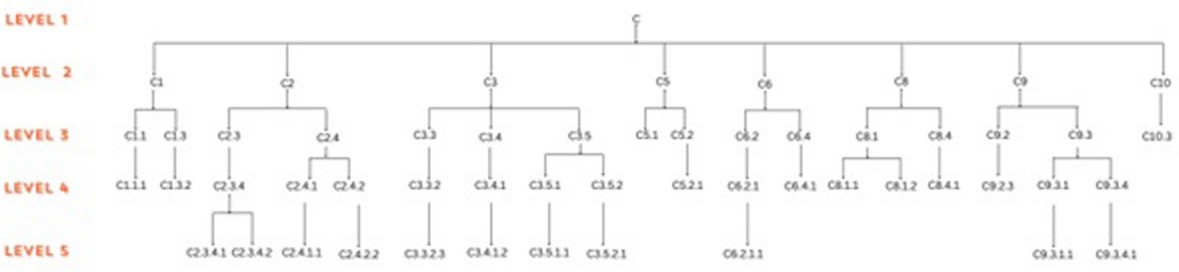

**Fig 5. Tracking down the presence of the fundamental genes within different Modules at different level of the network.**

**Table 1. Network breaking mechanism to understand the gene names and their sub-modules.**

| Network decomposition outline | Gene names in motifs |
|---|---|
| C→C2→C2.3→C2.3.4→C2.3.4.1 | *F10, PLAT, F2R* |
| C→C2→C2.3→C2.3.4→C2.3.4.2 | *MMP1, CTSG, F2RL1* |
| C→C2→C2.4→C2.4.1→C2.4.1.1 | *PDGFA, FGF13, PTN* |
| C→C2→C2.4→C2.4.2→C2.4.2.2 | *APOC2, SDC1, SDC4* |
| C→C3→C3.3→C3.3.2→C3.3.2.3 | *AP2B1, AP1S3, DNM2* |
| C→C3→C3.4→C3.4.1→C3.4.1.2 | *EGFL7, RASIP1, SOX18* |
| C→C3→C3.5→C3.5.1→C3.5.1.1 | *CLDN2, CLDN7, CDH3* |
| C→C3→C3.5→C3.5.2→C3.5.2.1 | *CLDN5, TJP1, PCDH1* |
| C→C6→C6.2→C6.2.1→C6.2.1.1 | *RNF213, EPSTI1, XAF1* |
| C→C9→C9.3→C9.3.1→C9.3.1.1 | *VIP, GAL, GNRH1* |
| C→C9→C9.3→C9.3.4→C9.3.4.1 | *NMU, VIPR2, GNG11* |

in the network, contributing to serving as the backbone of the network and functioning "**Table 2**".

Furthermore, gene-drug interaction from DGIdb were retrieved and listed in "Table 3".

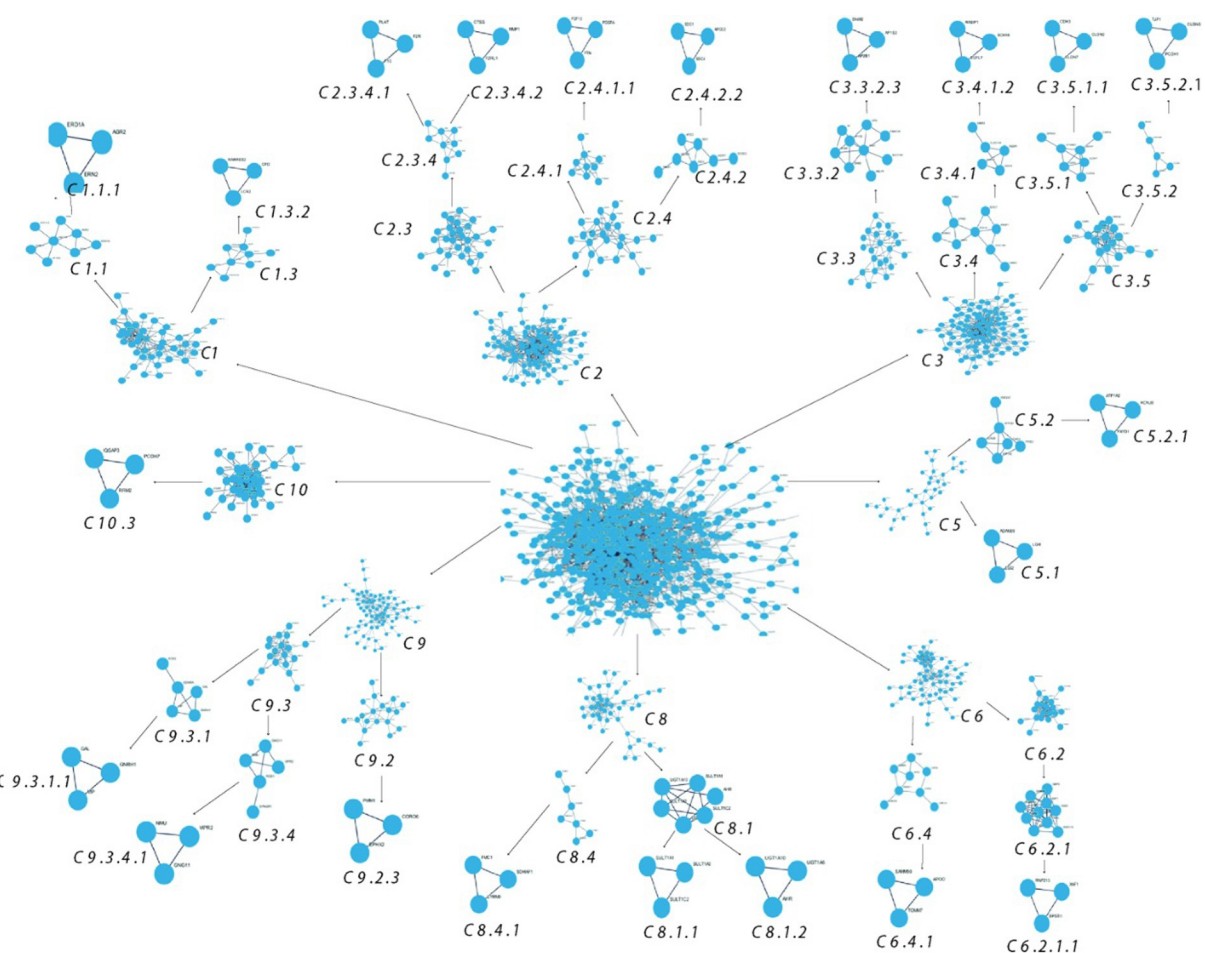

**Fig 6. Detecting communities within PDAC networks using leading eigenvector (LEV) method.**

Next, transcription factors were identified with TargetScan and listed in "Table 4".
Using miRTarBase, microRNAs were identified as listed in "Table 5".

## Discussion

The scarcity of knowledge regarding the genetic origins of PDAC necessitates further investigation into its genetic aspects. Despite the identification of several candidate genes for PDAC in recent years, the underlying mechanism responsible for PDAC development remains unclear. To the best of our knowledge, our in-silico study constitutes the initial endeavor to investigate the Protein-Protein Interaction (PPI) network of PDAC and to explore the varying contributions of the proteins encoded by the candidate genes in regulating the entire network through topological analysis.

Our investigation of the topological properties of the initial PDAC network, comprising 605 nodes and 2698 edges, reveals a weak hierarchical and scale-free fractal structure. The Girvan Newman algorithm helps us to identify a two-tier organization of the network, with one level representing local clustering of mostly low-degree nodes into well-defined successive communities or modules, and the other level representing more global connectivity in which hubs serve as higher-order communication points between interconnected communities. The fractal state of the network signifies self-similar organization, while the scale-free nature contributes to network stability. These topological properties facilitate efficient information processing within the network.

The objective of predicting candidate genes for diseases, such as exploring the role of gene interactions, is a fundamental goal of the medical sciences and is crucial for effective treatment. To achieve this, it is necessary to conduct preliminary analysis to identify potential biomarkers. In this study, we applied differential expression analysis on RNA-seq data of PDAC to identify transcriptomic signatures that are characteristic of the disease, and then performed network analysis to better understand the interactions between genes. The results of this study can be used to identify potential biomarkers for PDAC and contribute to the field of pharmacogenomics, which has significant applications in drug discovery. We focused on network-regulated genes in this study and found that the network of classified genes from PDAC displays hierarchical characteristics, indicating that the network is organized at the sub-module level. This hierarchical nature of the network is important for understanding the functional regulation of the disease. Individual gene activities are less important in this process.

Our networks, which comprise both up- and down-regulated genes, have led to the identification of 33 crucial regulators, including ($F10^{\downarrow}$, $PLAT^{\uparrow}$, $F2R^{\uparrow}$, $MMP1^{\uparrow}$, $CTSG^{\downarrow}$, $F2RL1^{\downarrow}$, $PDGFA^{\downarrow}$, $FGF13^{\downarrow}$, $PTN^{\downarrow}$, $APOC2^{\uparrow}$, $SDC1^{\uparrow}$, $SDC4^{\uparrow}$, $AP2B1^{\uparrow}$, $AP1S3^{\uparrow}$, $DNM2^{\uparrow}$, $EGFL7^{\downarrow}$, $RASIP1^{\downarrow}$,

**Table 2. Topological statistical properties of low degree node accompanied by high degree node as key regulators.**

| S.N. | Gene name | Degree | Betweenness Centrality | Closeness Centrality | Clustering Coefficient |
|------|-----------|--------|------------------------|----------------------|------------------------|
| 1 | F2RL1 | 4 | 0.0007 | 0.2791 | 0.6667 |
| 2 | F10 | 4 | 0.0008 | 0.3041 | 0.1666 |
| 3 | APOC2 | 8 | 0.0066 | 0.3030 | 0.2142 |
| 4 | CLDN2 | 11 | 0.0009 | 0.3351 | 0.8727 |
| 5 | PCDH1 | 3 | 0.0001 | 0.2835 | 0.3334 |
| 6 | MMP1 | 36 | 0.0241 | 0.3742 | 0.2730 |
| 7 | PLAT | 17 | 0.0091 | 0.3517 | 0.2332 |
| 8 | SDC1 | 40 | 0.0218 | 0.3735 | 0.2846 |
| 9 | TJP1 | 44 | 0.0275 | 0.3796 | 0.1987 |

**Table 3. Gene-drug interaction of the key regulators.**

| S. N. | Gene | Drug |
|---|---|---|
| 1 | F10 | EDOXABAN, IDRAPARINUX SODIUM, APIXABAN, RIVAROXABAN, HEPARIN, EDOXABAN TOSYLATE, EMICIZUMAB, BETRIXABAN, LETAXABAN, MENADIONE, OTAMIXABAN, IDRAPARINUX, TANOGITRAN, MELAGATRAN, TINZAPARIN, DANAPAROID, CHEMBL1271162, SEMULOPARIN, MANGIFERIN HEPTASULFATE, THROMBIN, TRIMETHOPRIM/SULFADOXINE |
| 2 | PLAT | AMINOCAPROIC ACID, ATORVASTATIN, MELPHALAN, EPOETIN BETA, RALOXIFENE, NAPROXEN, UROKINASE, BORTEZOMIB |
| 3 | F2R | RIGOSERTIB SODIUM, ATOPAXAR, VORAPAXAR SULFATE, VORAPAXAR, ATROPINE, WORTMANNIN, ARGATROBAN, BLEOMYCIN, LEPIRUDIN, THALIDOMIDE, MORPHINE, ALCOHOL, ASPIRIN, THROMBIN, DALTEPARIN, RUSALATIDE |
| 4 | MMP1 | DOXYCYCLINE CALCIUM, APRATASTAT, DOXYCYCLINE, DOXYCYCLINE HYCLATE, CIPEMASTAT, MARIMASTAT, PRINOMASTAT, LEUPROLIDE ACETATE, SIROLIMUS, COLLAGENASE CLOSTRIDIUM HISTOLYTICUM, MEDROXYPROGESTERONE ACETATE, LAMIVUDINE, LEFLUNOMIDE, HYDROCORTISONE, PENTOSAN POLYSULFATE SODIUM, TRIAMCINOLONE, RIBAVIRIN |
| 5 | CTSG | MANNITOL, CHEMBL374027 |
| 6 | F2RL1 | CHEMBL493076, CHEMBL494502, ROXITHROMYCIN, MINOCYCLINE, TETRACYCLINE, ERYTHROMYCIN, CLARITHROMYCIN, DOXYCYCLINE |
| 7 | PDGFA | SUNITINIB, SQUALAMINE |
| 8 | SDC1 | HEPARIN, INDATUXIMAB RAVTANSINE |
| 9 | EGFL7 | PARSATUZUMAB |
| 10 | CDH3 | PF-06671008 |
| 11 | TJP1 | RISPERIDONE, GENISTEIN, ALCOHOL, DEXAMETHASONE |
| 12 | VIP | DIGOXIN, LISINOPRIL, OMEPRAZOLE, RIBAVIRIN, ANDROSTANOLONE, AZASERINE, FLUTAMIDE |
| 13 | GAL | LIOTHYRONINE SODIUM, HYDROCORTISONE, PRASTERONE, DIACETYLMORPHINE, MIRTAZAPINE |
| 14 | GNRH1 | RESERPINE, RALOXIFENE, GOSERELIN, AMINOGLUTETHIMIDE, ZALCITABINE, DAPSONE, CAPTOPRIL, CHEMBL208519, LITHIUM, LEUPROLIDE, DITIOCARB |

$SOX18^{\downarrow}$, $CLDN2^{\uparrow}$, $CLDN7^{\uparrow}$, $CDH3^{\uparrow}$, $CLDN5^{\downarrow}$, $TJP1^{\uparrow}$, $PCDH1^{\downarrow}$, $RNF213^{\uparrow}$, $EPSTI1^{\uparrow}$, $XAF1^{\uparrow}$, $VIP^{\downarrow}$, $GAL^{\downarrow}$, $GNRH1^{\downarrow}$, $NMU^{\uparrow}$, $VIPR2^{\downarrow}$, and $GNG11^{\downarrow}$). These regulators were identified through the analysis of motifs and module regulation, and their biological importance, roles in network activities and associated regulations, and potential as targets for disease have been established. Additionally, the biological activities and pathways in which these key regulator genes are involved have been identified.

Studies have shown the strong potential of F10 to improve treatment outcomes in acute myeloid leukemia, acute lymphocytic leukemia, glioblastoma, and prostate cancer [47]. The PLAT gene has been studied in the context of cancer in several papers. PLAT, also known as tissue-type plasminogen activator, has been found to play a role in gefitinib resistance in non-small cell lung cancer (NSCLC) [48]. Mutations and alterations in the FGFR2 gene have been found to play a significant role in the development and progression of various types of cancer, including endometrial cancer, breast cancer, and gastrointestinal/genitourinary tract cancers. These alterations include somatic hotspot mutations, structural amplifications, and fusions [49]. Pleiotrophin (PTN) is a gene that has been found to be differentially expressed in various types of cancer, including hepatocellular carcinoma (HCC) [50], oral squamous cell carcinoma (OSCC), ovarian cancer, and breast cancer (BrCa). Tight junction proteins ZO-1, TJP1, TJP2, and TJP3 are scaffolding proteins that connect trans-membrane proteins like claudins and occludin to the actin cytoskeleton. They play a crucial role in maintaining the integrity of tight

**Table 4. Transcription factor of Key regulator genes.**

| S.N. | Key TF | Description | P-value | Key regulator Genes |
|---|---|---|---|---|
| 1 | NF1 | neurofibromin 1 | 0.000266 | GNRH1,PLAT |
| 2 | SP3 | Sp3 transcription factor | 0.001 | F2R,PLAT,SOX18 |
| 3 | SP1 | Sp1 transcription factor | 0.00127 | CDH3,F10,PLAT,PTN,F2R |
| 4 | TWIST1 | twist basic helix-loop-helix transcription factor 1 | 0.0017 | F2R,MMP1 |
| 5 | MYB | v-myb myeloblastosis viral oncogene homolog (avian) | 0.0019 | NMU,CTSG |
| 6 | GATA3 | GATA binding protein 3 | 0.002 | CDH3,MMP1 |
| 7 | POU2F1 | POU class 2 homeobox 1 | 0.002 | GNRH1,SDC4 |
| 8 | JUN | jun proto-oncogene | 0.00221 | PLAT,PTN,MMP1 |
| 9 | PPARG | peroxisome proliferator-activated receptor gamma | 0.00593 | MMP1,CLDN2 |
| 10 | HDAC1 | histone deacetylase 1 | 0.00684 | TJP1,CLDN7 |
| 11 | STAT1 | signal transducer and activator of transcription 1, 91kDa | 0.00945 | XAF1,VIP |
| 12 | CREB1 | cAMP responsive element binding protein 1 | 0.0108 | VIP,PLAT |
| 13 | STAT3 | signal transducer and activator of transcription 3 (acute-phase response factor) | 0.0255 | F2R,MMP1 |
| 14 | RELA | v-rel reticuloendotheliosis viral oncogene homolog A (avian) | 0.097 | SDC1,MMP1 |
| 15 | NFKB1 | nuclear factor of kappa light polypeptide gene enhancer in B-cells 1 | 0.0981 | SDC1,MMP1 |

junctions and regulating para-cellular permeability [51]. F2RL1, also known as Fc receptor-like 2, has been studied in the context of cancer. In metastatic breast cancer, decreased expression of FCRL2 mRNA was observed in brain metastatic tissues compared to primary breast tumors, and its expression in primary tumors was correlated with patient survival [52]. PDGFA is a protein that has been implicated in cancer initiation and progression. It is up-regulated in several cancers, including colorectal cancer (CRC) [53]. FGF13 has been found to play a role in cancer progression and treatment resistance. It has been shown to be associated with tumor growth and metastasis in pancreatic cancer [54]. In human pluripotent stem cells, disrupting TJP1 leads to the activation of bone morphogenic protein-4 (BMP4) signaling and loss of patterning phenotype [55]. CTSG has also been implicated in triple-negative breast cancer (TNBC), where it is overexpressed and correlated with a poor prognosis. CTGF, a protein that binds to CTSG, activates the FAK/Src/NF-κB p65 signaling axis, resulting in the up-regulation of Glut3 and enhanced aerobic glycolysis in TNBC cells [56]. APOC2 has been identified as a potential diagnostic biomarker for cancer detection and as an auxiliary prognostic marker or marker for immunotherapy in certain tumor types [57]. SDC1 has been shown to play a tumor-suppressor role in CRCs [58]. In cervical cancer, SDC1 expression is associated with low differentiation and increased lymph node metastases [59]. High SDC1 expression in cervical cancer is also correlated with a poor prognosis [60]. The AP2B1 gene has been studied in various types of cancer. In lung cancer, the transcription factors AP2A and AP2B were found

**Table 5. miRTarBase scanned microRNAs with respect to key regulators.**

| S.N. | microRNA | Target Gene | Number of interactions | FDR | p-value |
|---|---|---|---|---|---|
| 1 | hsa-miR-1-3p | RNF213, F2RL1, AP1S3 | 4 | 0.619 | 0.0885 |
| 2 | hsa-miR-335-5p | F10, RNF213, VIP, CDH3, PLAT, CLDN7, XAF1, RASIP1, SOX18 | 9 | 0.619 | 0.043 |
| 3 | hsa-miR-526b-3p | MMP1, F2RL1, F2R | 3 | 0.619 | 0.114 |
| 4 | hsa-miR-615-3p | TJP1, DNM2, AP2B1 | 3 | 0.619 | 0.229 |
| 5 | hsa-miR-106b-5p | AP2B1, F2R, F2RL1 | 3 | 0.626 | 0.331 |
| 6 | hsa-miR-17-5p | F2RL1, F2R, TJP1 | 3 | 0.641 | 0.378 |
| 7 | hsa-miR-124-3p | DNM2, SDC4, RASIP1 | 3 | 0.666 | 0.513 |

to promote the expression of the USP22 gene, which is associated with aggressive growth and therapy resistance [61]. Patients with higher AP1S1 expression have higher estrogen receptor gene expression, increased risk of distant metastasis and lymph node metastasis, and worse overall survival rates [62]. DNM2 appears to be a promising molecular target for the development of anti-invasive agents and has shown potential in reducing cell proliferation and inducing apoptosis in cancer cells [63].

EGFL7 is a gene that has been found to play a role in cancer. It has been identified as a driver gene for resistance to EGFR kinase inhibition in lung cancer cells [64]. In addition, RASIP1 is negatively regulated by fork-head box O3 (FOXO3), which suppresses DLBCL cell proliferation. FOXO3 binds to the promoter sequence of RASIP1 and inhibits its transcription [65]. In lung cancer, the inhibition of SOX18 with a specific inhibitor called Sm4 has shown cytotoxic effects on non-small cell lung cancer (NSCLC) cell lines, leading to cell cycle disruption and up-regulation of p21, a key regulator of cell cycle progression [66]. In various cancers, including ovarian, testicular, endocervical, liver, and lung adenocarcinoma, CLDN7 is highly expressed and activates multiple signaling pathways involved in tumor growth, migration, invasion, and chemo-resistance [67]. In OSCC, CDH3 is up-regulated and associated with a poor prognosis, promoting migration, invasion, and chemo-resistance in oral squamous cell carcinoma [68]. CLDN5 expression levels differ significantly between cancer and normal tissues, and it has been confirmed in multiple studies [69]. CLDN5 is implicated in the oncogenesis of diverse cancer types, highlighting its potential significance in cancer biology [70]. PCDH1 enhances p65 nuclear localization by interacting with KPNB1, activating the NF-κB signaling pathway and promoting PDAC progression [71]. PCDH1 can be used as a negative prognostic marker and a potential therapeutic target for PDAC patients [72]. In breast cancer, RNF213 is differentially expressed in primary tumors and is correlated with overall survival in patients with basal-like sub-type breast cancer [73]. Additionally, RNF213 knockdown disrupts angiogenesis and sensitizes endothelial cells to inflammation, leading to altered angiogenesis and potential links to Moyamoya disease. VIP expression has been associated with the transcription factor ZEB1, which is known to regulate EMT [74]. In breast cancer, VIP receptor 2 (VIPR2) has been shown to promote cell proliferation and tumor growth through the cAMP/PKA/ERK signaling pathway [75]. EPSTI1 interacts with valosin-containing protein to activate nuclear factor κ-light-chain-enhancer of activated B cells (NF-κB) and inhibit apoptosis [76]. EPSTI1 has also been implicated in immune response, as it promotes the expression of viral response genes and is associated with immune privilege and autoimmune diseases [77]. NMU is expressed at higher levels in tumor tissues compared to normal tissues, and its expression has been associated with poor prognosis and shorter overall survival in cancer patients [78]. Studies have shown that VIPR2 overexpression promotes cell proliferation in breast cancer cell lines and exacerbates tumor growth in vivo [75]. Furthermore, VIPR2 has been shown to form homodimers and oligomers, which are involved in VIP-induced cancer cell migration [79]. The expression of GNG11 mRNA is down-regulated in ovarian cancer patients, and its high expression is associated with poor prognosis [80]. GNG11 may play a crucial role in the biological process of ovarian cancer through the ECM-receptor interaction pathway [81].

The study showing the role of proteinase-activated receptor 2 (PAR2) in TGF-β1-dependent cell motility underlines the importance of PAR2 in the tumor microenvironment. The proteinase-activated receptor 2 (PAR2) is crucial for TGF-β1-dependent cell motility, establishing PAR2 as a key player in PDAC invasion and metastasis [82]. PAR2 is a G protein-coupled receptor (GPCR) that senses extracellular proteases, particularly serine proteinases, which are abundant in the PDAC microenvironment. The interaction between PAR2 and TGF-β1 promotes tumor cell movement, a process essential for invasion into surrounding tissues and metastasis, which are hallmarks of PDAC. High expression of TGF-β1 further contributes to

immune evasion and the development of a dense stromal environment, which creates barriers to effective therapy. TGF-β1 is well-known for its dual role in cancer, functioning as a tumor suppressor in early-stage cancers and as a promoter of invasion and metastasis in later stages, particularly in PDAC. The cooperation between cadherin-1 (E-cadherin) and cadherin-3 (P-cadherin), both critical cell adhesion molecules, plays a pivotal role in determining PDAC aggressiveness [83]. While E-cadherin downregulation promotes epithelial-mesenchymal transition (EMT), a key process in cancer metastasis, P-cadherin overexpression in PDAC may exacerbate this effect, leading to increased tumor invasiveness. This interplay highlights the molecular dysregulation of adhesion mechanisms in PDAC, facilitating the cancer's spread to distant sites. The research showing that 92% of PDAC cases had positive immunostaining for claudin-4 and 58% for claudin-1 [84] suggests that claudins are critical markers in PDAC pathology. Claudins are integral components of tight junctions, which regulate paracellular permeability and maintain epithelial barrier function. In PDAC, the overexpression of claudins, particularly claudin-4, has been associated with tumor invasion and metastasis. These finding positions claudin-4 as a potential biomarker for PDAC, with therapeutic implications given its overexpression in a majority of PDAC tumors. Claudin-4 may also influence the permeability of the tumor microenvironment, facilitating cancer cell dissemination. The study by T. Ito et al. [85] focused on matrix metalloproteinase-1 (MMP-1), which plays a crucial role in tissue remodeling and degradation of the extracellular matrix (ECM). MMP-1 is frequently upregulated in cancer and contributes to tumor invasion and metastasis by breaking down the ECM barriers that normally confine cells. In the analysis of PDAC tissues, MMP-1 was found to be significantly elevated, suggesting its involvement in PDAC progression, particularly in enabling tumor cells to invade surrounding tissues. This highlights MMP-1 as a potential therapeutic target, where inhibiting its activity might reduce the invasive capabilities of PDAC cells. Additionally, neuromedin U (NmU) has been implicated in PDAC pathogenesis, providing novel insights into its role in cancer biology [86]. NmU, a neuropeptide known for its role in immune response and regulation of smooth muscle contraction, appears to promote tumor growth and metastasis in PDAC. The upregulation of NmU in PDAC tissues correlates with more aggressive tumor behavior, suggesting that NmU-targeted therapies could offer new avenues for treating PDAC, particularly in cases where NmU expression is high. Another significant finding in PDAC research is the role of microRNA miR-7-5p, which acts as a tumor suppressor by targeting SOX18, a transcription factor involved in angiogenesis and EMT [87]. The downregulation of miR-7-5p in PDAC leads to SOX18 overexpression, promoting tumor progression through enhanced angiogenesis and increased metastatic potential. Restoring miR-7-5p levels could inhibit tumor growth and reduce metastasis, offering potential therapeutic strategies focused on microRNA-based therapies in PDAC.

The neuroactive ligand-receptor interaction pathway primarily involves neurotransmitters and their receptors, which traditionally mediate signaling in the nervous system. However, increasing evidence indicates that these molecules also have significant roles in tumor biology, including PDAC. Several neurotransmitters, such as serotonin, dopamine, and norepinephrine, are known to interact with their respective receptors on pancreatic cancer cells, influencing signaling pathways that promote tumor cell survival, proliferation, and metastasis. For example: Serotonin receptors (5-HT receptors) have been found to be overexpressed in PDAC, enhancing tumor growth. Serotonin signaling activates downstream oncogenic pathways such as the PI3K/AKT and RAS/ERK pathways, which promote cell proliferation and survival. Inhibition of these receptors or signaling pathways has shown to reduce PDAC cell proliferation, indicating a direct role of neurotransmitter-receptor interactions in the tumor's growth dynamics [88]. One of the hallmark features of PDAC is perineural invasion (PNI), where cancer cells spread along nerve fibers. Neuroactive ligand-receptor interactions are

central to this process, as tumor cells communicate with nerve cells to enhance their migration along these neural pathways. Perineural invasion in PDAC correlates with worse prognosis and increased metastasis, especially since neural invasion facilitates access to distant organs [89]. Cancer cells exploit neurotransmitter signaling to promote their affinity for nerves. This invasion is mediated by neurotrophic factors, such as nerve growth factor (NGF) and glial cell-derived neurotrophic factor (GDNF), which bind to their receptors (TrkA and RET, respectively) on PDAC cells, promoting invasion toward and along nerves [90]. Neuroactive ligand-receptor signaling also has implications for cancer-related pain, a common and severe symptom in PDAC patients. Pancreatic tumors modulate pain receptors (nociceptors) through the release of certain neurotransmitters and inflammatory molecules, exacerbating the experience of pain in patients. This is often due to the activation of the transient receptor potential vanilloid 1 (TRPV1) and other pain-related receptors expressed in pancreatic nerve fibers, which become hyperactivated as a result of neuro-ligand interactions. The persistent stimulation of these receptors amplifies pain signals to the brain. Neurotransmitter signaling also contributes to immune evasion. For instance, dopamine receptors on immune cells can suppress the immune response by inhibiting the activity of cytotoxic T-cells and promoting the recruitment of regulatory T-cells (Tregs), both of which support tumor immune escape [91].

The leukocyte transendothelial migration pathway refers to the movement of leukocytes (immune cells) across the endothelial barrier from the bloodstream into tissue, typically during immune surveillance or inflammation. However, in PDAC, this pathway is co-opted by the tumor to support its progression. PDAC is characterized by a highly immunosuppressive tumor microenvironment (TME), in part due to altered leukocyte migration. While immune cells, such as macrophages, neutrophils, and T-cells, migrate into the tumor, they are often reprogrammed by the tumor to support tumor growth rather than launch an anti-tumor immune response. Tumor-associated macrophages (TAMs), which migrate through this pathway, are frequently polarized into the M2 phenotype, which supports tumor growth, suppresses inflammation, and promotes tissue remodeling. The M2 macrophages secrete immunosuppressive cytokines like IL-10 and TGF-β, which inhibit effective anti-tumor immune responses. Regulatory T-cells (Tregs), which also migrate into the tumor via transendothelial migration, inhibit the activity of cytotoxic T-cells and natural killer (NK) cells, further reducing the immune system's ability to attack the tumor. This leads to immune evasion, a hallmark of PDAC, where the tumor successfully suppresses immune surveillance mechanisms. The interaction between leukocytes and endothelial cells during migration is not only important for immune cell infiltration but also contributes to tumor angiogenesis and metastasis. Leukocytes, particularly TAMs, secrete factors like vascular endothelial growth factor (VEGF) and matrix metalloproteinases (MMPs), which promote angiogenesis and metastasis [92]. Leukocytes migrating into the PDAC tumor microenvironment contribute to a chronic inflammatory state that paradoxically promotes tumor growth. While leukocyte infiltration typically signifies an immune response, in PDAC, the tumor co-opts this inflammatory response to sustain its own progression. The inflammatory cytokines secreted by leukocytes, such as TNF-α, IL-6, and IL-1β, promote cancer cell proliferation, survival, and resistance to apoptosis. Chronic inflammation within the PDAC microenvironment is associated with genetic instability, increased mutation rates, and the activation of oncogenic pathways, such as the NF-κB and STAT3 pathways, further accelerating tumor progression [93].

Both the neuroactive ligand-receptor interaction and leukocyte transendothelial migration pathways are intricately linked to the aggressive phenotype of PDAC. Their roles extend beyond simple signaling processes to become central to the immune suppression, perineural invasion, metastasis, and tumor-promoting inflammation that characterize PDAC(92). Neuroactive ligand-receptor interactions support tumor growth by activating oncogenic signaling,

promoting nerve invasion, and contributing to cancer-related symptoms like pain, all of which are directly linked to the poor prognosis seen in PDAC patients (93). Leukocyte transendothelial migration, on the other hand, is key in shaping the immunosuppressive microenvironment, promoting angiogenesis, and aiding in metastasis, which are central to PDAC's resistance to treatment and its high metastatic potential [92].

Furthermore, key regulator genes were enriched in G-protein beta-sub-unit binding, receptor binding, serine-type endopeptidase activity and growth factor activity as molecular function. While important cellular process are extracellular region, plasma membrane, and cell surface. as listed in "Table 6". Go classification of key signature genes indicates the regulation of positive regulation of cell proliferation, positive regulation of protein kinase B signaling, calcium-independent cell-cell adhesion via plasma membrane cell-adhesion molecules related biological process. Next, the pathway crosstalk analysis explored the interactions among significantly enriched pathways. Genes VIPR2, GAL, F2R, NMU, F2RL1, GNRH1, CTSG, VIP were involved in neuroactive ligand-receptor interaction pathways. Key regulator genes CLDN5, CDH3, SDC4, CLDN7, SDC1, CLDN2 were active in Cell adhesion molecules and CLDN5, CLDN7, CLDN2 enriched in Leukocyte trans-endothelial migration pathway.

Further, gene-drug interaction for regulator genes retrieved drugs against 14 genes as listed in "Table 3". Using DGIdb, we conducted a query to identify interactions between a predefined list of genes (33 key regulator genes) implicated in pancreatic ductal adenocarcinoma and known drugs. The key regulator gene list was derived from this study, and DGIdb was queried with approved drugs parameters, capturing interaction types such as inhibition, activation, and binding. We identified 91 drug-gene interactions involving 14 unique genes and 79 approved distinct drugs. Among these, genes MMP1, F10, F2R, GNRH1 had the highest number of interactions, interacting with more than 20 different drugs. The interactions included various types such as inhibition (50%), activation (30%), and binding (20%). The identification of MMP1, F10, F2R, GNRH1 as a highly interactive gene underscores its potential role in PDAC.

We identified 15 key transcription factors, including NF1(neurofibromin 1), NFKB1 (nuclear factor of kappa light polypeptide gene enhancer in B-cells 1), and STAT1(signal transducer and activator of transcription 3 (acute-phase response factor)), that regulate our genes of

**Table 6. Enrichment analysis results for key signature genes.**

| Category | Description | Genes | P-value | Fold Enrichment |
|---|---|---|---|---|
| MF | G-protein beta-subunit binding | F2R, F2RL1, GNG11 | 0.000697 | 74.125 |
| MF | receptor binding | EGFL7, F2R, NMU, F2RL1, PLAT | 0.00433306 | 7.196601942 |
| MF | serine-type endopeptidase activity | F10, MMP1, CTSG, PLAT | 0.003455303 | 12.68449198 |
| MF | growth factor activity | PDGFA, PTN, FGF13 | 0.040571326 | 9.076530612 |
| CC | extracellular region | EGFL7, F10, MMP1, F2R, PDGFA, PLAT, PTN, GAL, APOC2, NMU, GNRH1, CTSG, VIP, FGF13 | 0.00000756 | 4.084387709 |
| CC | plasma membrane | VIPR2, F10, SDC4, F2R, AP2B1, PTN, CLDN2, GNG11, DNM2, TJP1, CLDN5, CDH3, CLDN7, F2RL1, SDC1, CTSG, PCDH1, FGF13 | 0.000728 | 2.129213998 |
| CC | cell surface | EGFL7, SDC4, F2R, PDGFA, SDC1, CTSG, PLAT | 0.00038 | 6.868351064 |
| BP | positive regulation of cell proliferation | TJP1, CLDN5, F2R, CLDN7, PDGFA, PTN, VIP | 0.00019 | 7.811582569 |
| BP | positive regulation of protein kinase B signaling | F10, F2R, PDGFA, F2RL1 | 0.003123059 | 13.15 |
| BP | calcium-independent cell-cell adhesion via plasma membrane cell-adhesion molecules | CLDN5, CLDN7, CLDN2 | 0.000506 | 86.88392857 |
| KEGG | Neuroactive ligand-receptor interaction | VIPR2, GAL, F2R, NMU, F2RL1, GNRH1, CTSG, VIP | 0.0000453 | 7.486430518 |
| KEGG | | CLDN5, CDH3, SDC4, CLDN7, SDC1, CLDN2 | 0.0000634 | 13.04202532 |

interest. as listed in "Table 4". Some well-known transcription factors among them are STAT1-signal transducer and activator of transcription 1, STAT3-signal transducer and activator of transcription 3 (acute-phase response factor), CREB1-cAMP responsive element binding protein 1, RELA-v-rel reticuloendotheliosis viral oncogene homolog A (avian),and NFKB1-nuclear factor of kappa light polypeptide gene enhancer in B-cells 1. Notably, NFKB1 was found to regulate a network of genes involved in positive regulation of protein kinase B signaling process or Cell adhesion molecules pathway, highlighting its potential role in PDAC.

MicroRNAs (miRNAs) are crucial post-transcriptional regulators that can influence gene expression by targeting transcription factors (TFs). In this study, we utilized the TARGET SCAN database to predict miRNAs that regulate TFs involved in pancreatic ductal adenocarcinoma. Next, using miRTarBase, we identified 7 key miRNAs predicted to target transcription factors, including hsa-miR-1-3p, hsa-miR-335-5p, hsa-miR-526b-3p, hsa-miR-106b-5p, hsa-miR-17-5p and hsa-miR-124-3p as listed in "Table 5". Notably, hsa-miR-335-5p was predicted to target 9 key genes with a high context score (**FDR** = 0.619, **p-value** = 0.043), suggesting a strong regulatory interaction. Additionally, hsa-miR-1-3p was predicted to target *RNF213*, *F2RL1*, *AP1S3*, indicating its potential role in regulating multiple pathways. The identification of hsa-miR-17-5p as a regulator of *F2RL1*, *F2R*, *TJP1* suggests a significant role in positive regulation of cell proliferation. The high context score (**FDR** = 0.641, **p-value** = 0.378) indicates a strong interaction, which could be critical for modulating positive regulation of cell proliferation.

In this study, we applied a systematic and extensible methodology, which finds significant key regulators in PDAC RNA-seq data. This study suggested that gene expression profiling can be used to differentiate and identify patients from their healthy counterparts. However, experimental validation of these results in large sample size would confirm the reliableness of these key regulators and facilitate the designing of an economical and susceptive molecular diagnostic. These findings warrant further investigation to validate the interactions and explore their clinical applications.

## Limitation of the study

Nevertheless, further investigation is necessary to verify the expression and function of the key regulators identified in PDAC, given the constrained sample size.

## Conclusion

We conducted a comprehensive analysis using one RNA-seq gene expression profiles, comparing individuals with PDAC to healthy controls. Our aim was to identify Differentially Expressed Genes (DEGs) and elucidate their biological functions through pathway enrichment analysis. Additionally, we explored the topological features of the gene interaction network, uncovering significant key regulators associated with PDAC. Moreover, we investigated approved drugs, transcription factors and mi-RNAs targeting these key regulators. These genes are anticipated to play a crucial role in PDAC progression. This study can be considered as very useful in the context of personalized treatment plan for PDAC patients. Personalized treatment in cancer represents a paradigm shift towards individualized and precision-based approaches to diagnosis, prognosis, and therapy selection, with the ultimate goal of improving patient outcomes and quality of life.

## Acknowledgments

We are thankful to the HPC of Ashoka University for providing the necessary infrastructure to help us successfully conduct our research work. We express our gratitude to Ram Nayan

Verma for his valuable and insightful contributions to the discussions on the network analysis method utilized in this study.

## Author Contributions

**Conceptualization:** Kankana Bhattacharjee, Aryya Ghosh.

**Data curation:** Kankana Bhattacharjee.

**Formal analysis:** Kankana Bhattacharjee.

**Investigation:** Aryya Ghosh.

**Supervision:** Aryya Ghosh.

**Writing – original draft:** Kankana Bhattacharjee.

**Writing – review & editing:** Aryya Ghosh.

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
