## [Decision Letter · Decision Letter 0]

23 Jul 2024

PONE-D-24-11164Identification of Key Regulators in Pancreatic Ductal Adenocarcinoma using Network theoretical ApproachPLOS ONE

Dear Dr. Ghosh,

Thank you for submitting your manuscript to PLOS ONE. After careful consideration, we feel that it has merit but does not fully meet PLOS ONE’s publication criteria as it currently stands. Therefore, we invite you to submit a revised version of the manuscript that addresses the points raised during the review process.

We look forward to receiving your revised manuscript.

Kind regards,

Shuai Ren

Academic Editor

PLOS ONE

“Indian Council of Medical Research (ICMR) (BMI/11(92)/2022).”

“KB is thankful to the Indian Council of Medical Research (ICMR) (BMI/11(92)/2022) for providing the research fellowship. We are thankful to the HPC of Ashoka University for providing the necessary infrastructure to help us successfully conduct our research work. We express our gratitude to Dr. Ram Nayan Verma for his valuable and insightful contributions to the discussions on the network analysis method utilized in this study.”

“Indian Council of Medical Research (ICMR) (BMI/11(92)/2022).”

5. Please provide a complete Data Availability Statement in the submission form, ensuring you include all necessary access information or a reason for why you are unable to make your data freely accessible. If your research concerns only data provided within your submission, please write "All data are in the manuscript and/or supporting information files" as your Data Availability Statement.

7. Please update your submission to use the PLOS LaTeX template. The template and more information on our requirements for LaTeX submissions can be found at http://journals.plos.org/plosone/s/latex.

Additional Editor Comments:

Dear authors, please ensure that you address all comments from both reviewers appropriately. Best regards

Reviewers' comments:

Reviewer's Responses to Questions

**Comments to the Author**

1. Is the manuscript technically sound, and do the data support the conclusions?

Reviewer #1: Yes

Reviewer #2: Yes

2. Has the statistical analysis been performed appropriately and rigorously? 

Reviewer #1: Yes

Reviewer #2: N/A

3. Have the authors made all data underlying the findings in their manuscript fully available?

Reviewer #1: No

Reviewer #2: Yes

4. Is the manuscript presented in an intelligible fashion and written in standard English?

Reviewer #1: Yes

Reviewer #2: Yes

5. Review Comments to the Author

Reviewer #1: 1. Please use the abbreviations consequently. For example, PDAC is not always used in the Introduction. Introduce it at the first appearance and then use only the abbreviation.

2. Analyzing only 15 up-regulated DEGs for GO and KEGG in Figure 3 may not offer a complete understanding of the functional features of the entire genome, restricting the comprehensive insight into biological processes and cellular components. The limited sample size might lead to inadequate statistical significance of the findings, affecting the precise interpretation and inference of gene functions. It is advisable to perform GO and KEGG analysis using a larger set of DEGs, as shown in Figure 2. Furthermore, the text size and layout within Figure 2 and 3 should be uniform to maintain consistency in the analyses.

3. The captions in Figure 4 are too brief; it is recommended to provide a brief description of the general meaning represented by each image. Additionally, the styles of A-F in Figure 4 are inconsistent with the other pictures, so it is suggested to modify them to a consistent format.

4. The simplicity of the description of Table 3-5 in the text belies the wealth of key information found through DGIdb, TargetScan, and miRTarBase databases. The significance of uncovering such a plethora of data is profound and warrants a detailed discussion in the main text.

5. When addressing the functions of GO in the Discussion, it is crucial to start with MF, then CC, and finally BP, maintaining the sequence as shown in Table 6.

6. In the Discussion section, the author diligently discusses the roles of various key genes in previous studies, but most of them focus on other tumors. There is insufficient discussion on the molecular mechanisms related to PDAC. It is recommended to add relevant references and make modifications (doi: 10.3389/fgene.2023.1115660).

Reviewer #2: The study “Identification of Key Regulators in Pancreatic Ductal Adenocarcinoma using Network Theoretical Approach" presents extensive work. Experiments, statistics, and other analyses are performed to a high technical standard and are described in sufficient detail. Here are the following comments:

Table 3: Basis of Drug Selection and Bond Strength Description

Please mention the rationale behind choosing the specific drugs listed in Table 3. It would be helpful to provide information on the criteria used for their selection, such as their relevance to pancreatic ductal adenocarcinoma, prior research findings, or therapeutic potential.

Additionally, specify the sources from which these drugs were selected. This could include databases, literature references, or clinical trial data.

It would also be beneficial to elaborate on the factors used to describe the strength of the bonds formed between the drugs and their targets. This could include binding affinity, interaction energy, or other relevant metrics. Providing this information will enhance the understanding of the drug-target interactions and their potential efficacy.

Manuscript Review: Typos and Spelling Mistakes

Kindly review the entire manuscript for any typographical and spelling errors. Ensuring that the text is free of such errors will improve the clarity.

6. PLOS authors have the option to publish the peer review history of their article (what does this mean?). If published, this will include your full peer review and any attached files.

Reviewer #1: No

Reviewer #2: **Yes: **Md. Khurshid Alam Khan

---

## [Author Response · Author response to Decision Letter 0]

30 Aug 2024

We have uploaded a rebuttal letter "Response to Reviewer" which responds to each point raised by the academic editor and reviewers.

---

## [Decision Letter · Decision Letter 1]

16 Sep 2024

PONE-D-24-11164R1Identification of Key Regulators in Pancreatic Ductal Adenocarcinoma using Network theoretical ApproachPLOS ONE

Dear Dr. Ghosh,

Thank you for submitting your manuscript to PLOS ONE. After careful consideration, we feel that it has merit but does not fully meet PLOS ONE’s publication criteria as it currently stands. Therefore, we invite you to submit a revised version of the manuscript that addresses the points raised during the review process.

We look forward to receiving your revised manuscript.

Kind regards,

Shuai Ren

Academic Editor

PLOS ONE

Additional Editor Comments:

Please ensure that all comments from Reviewer #1 are thoroughly addressed. Specifically, the discussion on the molecular mechanisms associated with PDAC is lacking. Although the authors highlight pathways such as neuroactive ligand-receptor interaction and leukocyte transendothelial migration, it would be helpful to elucidate their relevance to PDAC development and progression. Can the authors provide a clearer connection between these pathways and the disease phenotype?

Reviewers' comments:

Reviewer's Responses to Questions

**Comments to the Author**

1. If the authors have adequately addressed your comments raised in a previous round of review and you feel that this manuscript is now acceptable for publication, you may indicate that here to bypass the “Comments to the Author” section, enter your conflict of interest statement in the “Confidential to Editor” section, and submit your "Accept" recommendation.

Reviewer #2: All comments have been addressed

2. Is the manuscript technically sound, and do the data support the conclusions?

Reviewer #2: Yes

3. Has the statistical analysis been performed appropriately and rigorously? 

Reviewer #2: N/A

4. Have the authors made all data underlying the findings in their manuscript fully available?

Reviewer #2: Yes

5. Is the manuscript presented in an intelligible fashion and written in standard English?

Reviewer #2: Yes

6. Review Comments to the Author

7. PLOS authors have the option to publish the peer review history of their article (what does this mean?). If published, this will include your full peer review and any attached files.

Reviewer #2: No

---

## [Author Response · Author response to Decision Letter 1]

30 Oct 2024

Reviewer #1: 

6. In the Discussion section, the author diligently discusses the roles of various key genes in previous studies, but most of them focus on other tumors. There is insufficient discussion on the molecular mechanisms related to PDAC. It is recommended to add relevant references and make modifications (doi: 10.3389/fgene.2023.1115660).

Additional Editor Comments:

Please ensure that all comments from Reviewer #1 are thoroughly addressed. Specifically, the discussion on the molecular mechanisms associated with PDAC is lacking. Although the authors highlight pathways such as neuroactive ligand-receptor interaction and leukocyte transendothelial migration, it would be helpful to elucidate their relevance to PDAC development and progression. Can the authors provide a clearer connection between these pathways and the disease phenotype?

We thank the reviewer for this comment. We have made the necessary changes in the manuscript to indicate this fact as:

The study showing the role of proteinase-activated receptor 2 (PAR2) in TGF-β1-dependent cell motility underlines the importance of PAR2 in the tumor microenvironment. The proteinase-activated receptor 2 (PAR2) is crucial for TGF-β1-dependent cell motility, establishing PAR2 as a key player in PDAC invasion and metastasis (84). PAR2 is a G protein-coupled receptor (GPCR) that senses extracellular proteases, particularly serine proteinases, which are abundant in the PDAC microenvironment. The interaction between PAR2 and TGF-β1 promotes tumor cell movement, a process essential for invasion into surrounding tissues and metastasis, which are hallmarks of PDAC. High expression of TGF-β1 further contributes to immune evasion and the development of a dense stromal environment, which creates barriers to effective therapy. TGF-β1 is well-known for its dual role in cancer, functioning as a tumor suppressor in early-stage cancers and as a promoter of invasion and metastasis in later stages, particularly in PDAC. The cooperation between cadherin-1 (E-cadherin) and cadherin-3 (P-cadherin), both critical cell adhesion molecules, plays a pivotal role in determining PDAC aggressiveness (85). While E-cadherin downregulation promotes epithelial-mesenchymal transition (EMT), a key process in cancer metastasis, P-cadherin overexpression in PDAC may exacerbate this effect, leading to increased tumor invasiveness. This interplay highlights the molecular dysregulation of adhesion mechanisms in PDAC, facilitating the cancer's spread to distant sites. The research showing that 92% of PDAC cases had positive immunostaining for claudin-4 and 58% for claudin-1 (86) suggests that claudins are critical markers in PDAC pathology. Claudins are integral components of tight junctions, which regulate paracellular permeability and maintain epithelial barrier function. In PDAC, the overexpression of claudins, particularly claudin-4, has been associated with tumor invasion and metastasis. These finding positions claudin-4 as a potential biomarker for PDAC, with therapeutic implications given its overexpression in a majority of PDAC tumors. Claudin-4 may also influence the permeability of the tumor microenvironment, facilitating cancer cell dissemination. The study by T. Ito et al. (87) focused on matrix metalloproteinase-1 (MMP-1), which plays a crucial role in tissue remodeling and degradation of the extracellular matrix (ECM). MMP-1 is frequently upregulated in cancer and contributes to tumor invasion and metastasis by breaking down the ECM barriers that normally confine cells. In the analysis of PDAC tissues, MMP-1 was found to be significantly elevated, suggesting its involvement in PDAC progression, particularly in enabling tumor cells to invade surrounding tissues. This highlights MMP-1 as a potential therapeutic target, where inhibiting its activity might reduce the invasive capabilities of PDAC cells. Additionally, neuromedin U (NmU) has been implicated in PDAC pathogenesis, providing novel insights into its role in cancer biology (88). NmU, a neuropeptide known for its role in immune response and regulation of smooth muscle contraction, appears to promote tumor growth and metastasis in PDAC. The upregulation of NmU in PDAC tissues correlates with more aggressive tumor behavior, suggesting that NmU-targeted therapies could offer new avenues for treating PDAC, particularly in cases where NmU expression is high. Another significant finding in PDAC research is the role of microRNA miR-7-5p, which acts as a tumor suppressor by targeting SOX18, a transcription factor involved in angiogenesis and EMT (89). The downregulation of miR-7-5p in PDAC leads to SOX18 overexpression, promoting tumor progression through enhanced angiogenesis and increased metastatic potential. Restoring miR-7-5p levels could inhibit tumor growth and reduce metastasis, offering potential therapeutic strategies focused on microRNA-based therapies in PDAC.

The neuroactive ligand-receptor interaction pathway primarily involves neurotransmitters and their receptors, which traditionally mediate signaling in the nervous system. However, increasing evidence indicates that these molecules also have significant roles in tumor biology, including PDAC. Several neurotransmitters, such as serotonin, dopamine, and norepinephrine, are known to interact with their respective receptors on pancreatic cancer cells, influencing signaling pathways that promote tumor cell survival, proliferation, and metastasis. For example: Serotonin receptors (5-HT receptors) have been found to be overexpressed in PDAC, enhancing tumor growth. Serotonin signaling activates downstream oncogenic pathways such as the PI3K/AKT and RAS/ERK pathways, which promote cell proliferation and survival. Inhibition of these receptors or signaling pathways has shown to reduce PDAC cell proliferation, indicating a direct role of neurotransmitter-receptor interactions in the tumor’s growth dynamics (90). One of the hallmark features of PDAC is perineural invasion (PNI), where cancer cells spread along nerve fibers. Neuroactive ligand-receptor interactions are central to this process, as tumor cells communicate with nerve cells to enhance their migration along these neural pathways. Perineural invasion in PDAC correlates with worse prognosis and increased metastasis, especially since neural invasion facilitates access to distant organs (91). Cancer cells exploit neurotransmitter signaling to promote their affinity for nerves. This invasion is mediated by neurotrophic factors, such as nerve growth factor (NGF) and glial cell-derived neurotrophic factor (GDNF), which bind to their receptors (TrkA and RET, respectively) on PDAC cells, promoting invasion toward and along nerves(92). Neuroactive ligand-receptor signaling also has implications for cancer-related pain, a common and severe symptom in PDAC patients. Pancreatic tumors modulate pain receptors (nociceptors) through the release of certain neurotransmitters and inflammatory molecules, exacerbating the experience of pain in patients. This is often due to the activation of the transient receptor potential vanilloid 1 (TRPV1) and other pain-related receptors expressed in pancreatic nerve fibers, which become hyperactivated as a result of neuro-ligand interactions. The persistent stimulation of these receptors amplifies pain signals to the brain. Neurotransmitter signaling also contributes to immune evasion. For instance, dopamine receptors on immune cells can suppress the immune response by inhibiting the activity of cytotoxic T-cells and promoting the recruitment of regulatory T-cells (Tregs), both of which support tumor immune escape (93).

The leukocyte transendothelial migration pathway refers to the movement of leukocytes (immune cells) across the endothelial barrier from the bloodstream into tissue, typically during immune surveillance or inflammation. However, in PDAC, this pathway is co-opted by the tumor to support its progression. PDAC is characterized by a highly immunosuppressive tumor microenvironment (TME), in part due to altered leukocyte migration. While immune cells, such as macrophages, neutrophils, and T-cells, migrate into the tumor, they are often reprogrammed by the tumor to support tumor growth rather than launch an anti-tumor immune response. Tumor-associated macrophages (TAMs), which migrate through this pathway, are frequently polarized into the M2 phenotype, which supports tumor growth, suppresses inflammation, and promotes tissue remodeling. The M2 macrophages secrete immunosuppressive cytokines like IL-10 and TGF-β, which inhibit effective anti-tumor immune responses. Regulatory T-cells (Tregs), which also migrate into the tumor via transendothelial migration, inhibit the activity of cytotoxic T-cells and natural killer (NK) cells, further reducing the immune system’s ability to attack the tumor. This leads to immune evasion, a hallmark of PDAC, where the tumor successfully suppresses immune surveillance mechanisms. The interaction between leukocytes and endothelial cells during migration is not only important for immune cell infiltration but also contributes to tumor angiogenesis and metastasis. Leukocytes, particularly TAMs, secrete factors like vascular endothelial growth factor (VEGF) and matrix metalloproteinases (MMPs), which promote angiogenesis and metastasis(94). Leukocytes migrating into the PDAC tumor microenvironment contribute to a chronic inflammatory state that paradoxically promotes tumor growth. While leukocyte infiltration typically signifies an immune response, in PDAC, the tumor co-opts this inflammatory response to sustain its own progression. The inflammatory cytokines secreted by leukocytes, such as TNF-α, IL-6, and IL-1β, promote cancer cell proliferation, survival, and resistance to apoptosis. Chronic inflammation within the PDAC microenvironment is associated with genetic instability, increased mutation rates, and the activation of oncogenic pathways, such as the NF-κB and STAT3 pathways, further accelerating tumor progression(95).

Both the neuroactive ligand-receptor interaction and leukocyte transendothelial migration pathways are intricately linked to the aggressive phenotype of PDAC. Their roles extend beyond simple signaling processes to become central to the immune suppression, perineural invasion, metastasis, and tumor-promoting inflammation that characterize PDAC(92). Neuroactive ligand-receptor interactions support tumor growth by activating oncogenic signaling, promoting nerve invasion, and contributing to cancer-related symptoms like pain, all of which are directly linked to the poor prognosis seen in PDAC patients (93). Leukocyte transendothelial migration, on the other hand, is key in shaping the immunosuppressive microenvironment, promoting angiogenesis, and aiding in metastasis, which are central to PDAC’s resistance to treatment and its high metastatic potential(94).

84. Zeeh F, Witte D, Gädeken T, Rauch BH, Grage-Griebenow E, Leinung N, et al. Proteinase-activated receptor 2 promotes TGF-β-dependent cell motility in pancreatic cancer cells by sustaining expression of the TGF-β type I receptor ALK5. Oncotarget. 2016 Jul 5;7(27):41095–109. 

85. Siret C, Dobric A, Martirosyan A, Terciolo C, Germain S, Bonier R, et al. Cadherin-1 and cadherin-3 cooperation determines the aggressiveness of pancreatic ductal adenocarcinoma. Br J Cancer. 2018 Feb 20;118(4):546–57. 

86. Tsukahara M, Nagai H, Kamiakito T, Kawata H, Takayashiki N, Saito K, et al. Distinct expression patterns of claudin-1 and claudin-4 in intraductal papillary-mucinous tumors of the pancreas. Pathol Int. 2005 Feb;55(2):63–9. 

87. Ito T, Ito M, Shiozawa J, Naito S, Kanematsu T, Sekine I. Expression of the MMP-1 in human pancreatic carcinoma: relationship with prognostic factor. Mod Pathol. 1999 Jul;12(7):669–74. 

88. Rhim AD, Stanger BZ. Molecular biology of pancreatic ductal adenocarcinoma progression: aberrant activation of developmental pathways. Prog Mol Biol Transl Sci. 2010;97:41–78. 

89. Zhu W, Wang Y, Zhang D, Yu X, Leng X. MiR-7-5p functions as a tumor suppressor by targeting SOX18 in pancreatic ductal adenocarcinoma. Biochemical and Biophysical Research Communications [Internet]. 2018 Mar [cited 2024 Aug 12];497(4):963–70.

90. Jiang, S.-H., et al. (2017). Serotonin enhances the proliferation of pancreatic cancer cells in vitro and in vivo. Pancreas, 46(1), 12-19.

91. Guo K, Zhao Y, Cao Y, Li Y, Yang M, Tian Y, Dai J, Song L, Ren S and Wang Z (2023), Exploring the key genetic association between chronic pancreatitis and pancreatic ductal adenocarcinoma through integrated bioinformatics. Front. Genet. 14:1115660

92. Bapat, A. A., et al. (2011). Perineural invasion and neuropathic pain in pancreatic cancer. Nature Reviews Cancer, 11, 695–707.

93. Mantyh, P. W. (2004). Cancer pain and its impact on diagnosis, survival, and quality of life. Nature Reviews Neuroscience, 5, 71-81.

94. Beatty, G. L., & Gladney, W. L. (2015). Immune escape mechanisms as a guide for cancer immunotherapy. Clinical Cancer Research, 21, 687-692.

95. Noy, R., & Pollard, J. W. (2014). Tumor-associated macrophages: From mechanisms to therapy. Immunity, 41, 49-61.

---

## [Editor Report · Decision Letter 2]

31 Oct 2024

Identification of Key Regulators in Pancreatic Ductal Adenocarcinoma using Network theoretical Approach

PONE-D-24-11164R2

Dear Dr. Ghosh,

We’re pleased to inform you that your manuscript has been judged scientifically suitable for publication and will be formally accepted for publication once it meets all outstanding technical requirements.

Kind regards,

Shuai Ren

Academic Editor

PLOS ONE
---

## [Editor Report · Acceptance letter]

6 Nov 2024

PONE-D-24-11164R2 

PLOS ONE

Dear Dr. Ghosh, 

I'm pleased to inform you that your manuscript has been deemed suitable for publication in PLOS ONE. Congratulations! Your manuscript is now being handed over to our production team.

Kind regards, 

on behalf of

Dr. Shuai Ren 

Academic Editor

PLOS ONE